# Rhinosinusitis as an Immune-Related Adverse Event: Clinical Characteristics, Management, and Prognostic Implications in Metastatic Melanoma Patients

**DOI:** 10.3390/cancers17142297

**Published:** 2025-07-10

**Authors:** Amalia Anastasopoulou, Aikaterini Gkoufa, Panagiotis Kouzis, Georgios Kyriakakis, Michail Belivanis, Georgia Sypsa, Spyridon Bouros, Helen Gogas, Panagiotis T. Diamantopoulos

**Affiliations:** 1First Department of Internal Medicine, National and Kapodistrian University of Athens, Laikon General Hospital, 11527 Athens, Greece; amanast@med.uoa.gr (A.A.); panagiotiskouzis@yahoo.gr (P.K.); kyrmanakos@hotmail.com (G.K.); spbouros@gmail.com (S.B.); hgogas@med.uoa.gr (H.G.); diamp@med.uoa.gr (P.T.D.); 2Department of Infectious Diseases, Oncology Unit of First Department of Internal Medicine, National and Kapodistrian University of Athens, Laikon General Hospital, 11527 Athens, Greece; 3Department of Radiology, Laikon General Hospital, 11527 Athens, Greece; mpelmike@yahoo.gr (M.B.); zetasypsa@med.uoa.gr (G.S.)

**Keywords:** immunotherapy, immune checkpoint inhibitors, immune-related adverse events, rhinosinusitis, melanoma

## Abstract

The present study focuses on one such underrecognized, immune-related adverse event—rhinosinusitis. Through a retrospective analysis of 304 melanoma patients treated with immune checkpoint inhibitors, we identified a significant association between imaging-confirmed rhinosinusitis and improved overall survival in the metastatic setting. Notably, rhinosinusitis occurred in over 20% of patients, was significantly linked to eosinophilia, and appeared most frequently with nivolumab monotherapy. Despite being largely asymptomatic and aseptic, its presence correlated with a doubling of median overall survival from ICI initiation. Our findings suggest that rhinosinusitis may serve as a novel biomarker of therapeutic response and favorable prognosis in metastatic melanoma patients undergoing ICI therapy. We believe this study offers a meaningful contribution to the growing field of immunotoxicology and may prompt further exploration of sinonasal inflammation in immunotherapy-treated populations.

## 1. Introduction

Melanoma represents a well-studied type of cancer in which the administration of immune checkpoint inhibitors (ICIs) has become a dominant and effective first-line treatment strategy through the regulation of immune responses and defense against cancer cells. However, T-cell activation by ICIs also leads to a growing number of immune-related adverse events (irAEs) [1]. With the increasing number of published studies, our understanding of the underlying mechanisms driving these events has expanded. However, many aspects remain to be explored, as irAEs do not affect all patients; their incidence varies across studies and different regimens, while some individuals develop multiple adverse events [2]. Rhinosinusitis has been primarily reported in case studies involving patients with various cancer types receiving ICIs and is increasingly recognized as a potential irAE rather than a conventional infectious or allergic condition. This distinction may be supported by several immunological and clinical observations, such as the lack of response to antibiotics, the absence of certain inflammatory biomarkers, the temporal association with ICI administration, and the responsiveness to topical or systemic steroid administration. In melanoma patients, there are only two recently published small retrospective studies addressing this emerging clinical condition [3,4]. Table 1 summarizes the current, limited existing literature, after a comprehensive review, addressing rhinosinusitis following immunotherapy in melanoma patients [4,5,6,7,8,9,10,11,12]. To further expand the understanding of the incidence, symptoms, diagnostic approaches, management, and prognostic implications of rhinosinusitis as an irAE, we conducted a retrospective analysis of melanoma patients treated with different types of ICIs.

## 2. Materials and Methods

We conducted a retrospective analysis of melanoma patients treated with ICIs at a tertiary university referral center to investigate the incidence of both symptomatic and asymptomatic, imaging-confirmed rhinosinusitis. Additionally, we examined the type of ICI regimens administered, the therapeutic approaches used for managing symptomatic cases, outcomes related to rhinosinusitis, and its prognostic implications.

### 2.1. Study Population

Adult patients with advanced or metastatic melanoma who received ICIs at a tertiary reference center for melanoma, between June 2015 and February 2023, with available head CT scans at baseline and after ICI initiation, were included in this analysis. For each patient, we collected and analyzed demographic information, baseline clinical and laboratory data, and comprehensive treatment details. All ICI-based regimens were recorded, including whether monotherapy or combination therapy was used, as well as treatment intent, duration, and any associated adverse events. Additionally, prognostic factors and survival outcomes were recorded and analyzed. The study was approved by the Institutional Review Board of Laikon General Hospital, Athens, Greece (Laikon General Hospital, Athens, Greece, IRB protocol number 67/25.01.21).

### 2.2. Rhinosinusitis Staging and Data Collection

The Harvard scoring system was used for grading rhinosinusitis, based on CT scan assessment [13]. This scale evaluates inflammation as follows: 0—normal (<2 mm mucosal thickening on any sinus wall), 1—all unilateral disease or anatomic abnormality, 2—bilateral disease limited to ethmoidal or maxillary sinuses, 3—bilateral disease involving at least one sphenoidal or frontal sinus, 4—pansinusitis. We recorded the emergence of rhinosinusitis during ICI treatment, as well as all accompanying clinical features. Associations between the presence of rhinosinusitis and prognostic parameters within the cohort were also evaluated.

### 2.3. Statistical Analysis

IBM SPSS statistics version 26 (IBM Corporation, New York, NY, USA) was used for the statistical analysis of the results. The Pearson Chi-Square test was used for testing associations between categorical variables and the Independent-Samples Mann–Whitney U test was used for testing between a categorical variable with two levels and continuous variables. Overall survival since first ICI administration (OS_ICI_) was defined as the time from the first administration of an ICI until death from any cause. Kaplan–Meier analysis was used to estimate OS_ICI_. The level of significance for all statistical tests was set at a probability value lower than 5% (*p* < 0.05).

## 3. Results

### 3.1. Baseline Patient Characteristics

A total of 350 patients were analyzed. All patients had baseline brain imaging, while 46/350 had no brain imaging after initiating ICIs and were censored. None of the patients included in the study had imaging findings compatible with rhinosinusitis before initiating ICIs. Of the remaining 304 patients, 64 (21.1%) presented after ICI initiation with imaging findings compatible with rhinosinusitis (mucosal thickening, opacification of the sinus, air-fluid level, cyst), recognized in one of the follow-up brain imaging. The median time of onset of rhinosinusitis was 2 months (range 1–7).

### 3.2. Rhinosinusitis Grading, Accompanying Symptoms, and Management

Figure 1a illustrates imaging findings consistent with rhinosinusitis, while Figure 1b shows comparative imaging of the paranasal sinuses obtained before and after initiation of immunotherapy, highlighting post-treatment changes. Figure 2 presents characteristic features of rhinosinusitis. Referring to the severity of the imaging findings, based on the Harvard scoring system for rhinosinusitis, there were 32 grade 1 cases (50.0%), 13 grade 2 cases (20.3%), 15 grade 3 cases (23.4%), and 4 grade 4 cases (6.3%). Only six (9.4%) cases were symptomatic, with fever, headache, facial pressure, and cough being the principal complaints (among them all cases of pansinusitis), and three of them were treated with antibiotics. Corticosteroids were administrated in 5/64 (7.8%) patients for decongestion. All symptomatic patients were advised to implement topical measures. The emergence of rhinosinusitis was associated (simultaneously or not) with other irAEs in 28 (9.2%) patients, while it was the sole irAE in 36 (11.8%). Out of the remaining 240 patients, 59 (19.4%) presented with other irAEs (without rhinosinusitis) and 181 (59.5%) did not experience any irAE. Finally, there was a strong association between the development of rhinosinusitis and the presence of eosinophilia. In fact, the analysis of 304 patients with available data showed that out of 251 patients without eosinophila, only 48 (17.1%) developed rhinosinusitis, while among 53 patients with eosinophilia, 21 (39.6%) developed rhinosinusitis (X^2^ (1, N = 304) = 13.3, *p* < 0.001).

### 3.3. Correlation with Immunotherapy Regimens

Out of 64 patients, 26 (40.6%) developed rhinosinusitis when under treatment with nivolumab, 5 under pembrolizumab, 6 under ipilimumab, 5 under atezolizumab, 9 under ipilimumab/nivolumab, and the remaining under several ICI combinations (NKTR/nivolumab, pembrolizumab/talimogene laherparepvec (T-vec), relatlimab/nivolumab, pembrolizumab/quavonlimab (MK1308). The vast majority of patients (51, 86.4%) developed sinusitis during their first course of ICIs, while 6 (10.2%) developed it during their second-line ICI, and only 2 (3.4%) after the third ICI treatment line.

Among 64 patients who were treated with nivolumab monotherapy at first line, 21 (32.8%) developed rhinosinusitis, while among 23 patients treated with pembrolizumab at first line, 7 (30.4%) developed rhinosinusitis. Among nine patients treated with ipilimumab monotherapy at first line, three (33.3%) developed rhinosinusitis. Finally, among 29 patients treated with the combination of ipilimumab and nivolumab at first line, 10 (34.5%) developed rhinosinusitis (Pearson Chi-Square 0.992).

### 3.4. Prognostic Correlations

Regarding the prognostic impact of rhinosinusitis, among patients treated with ICIs for advanced or metastatic melanoma, the presence of rhinosinusitis was associated with a longer median OS_ICI_ (33.3 months 95% CI, 10.5–56.1 vs. 15.4 months, 95% CI, 8.4–22.3 months for patients without imaging evidence of rhinosinusitis, *p* = 0.025). Among patients treated in the adjuvant setting, there was no effect of rhinosinusitis on OS_ICI_ (99.8 months vs. 113.0 months, 95% CI NR vs. 30.9–195.0 months for patients with and without rhinosinusitis, *p* = 0.439). There was no difference in the OS_ICI_ of patients with advanced or metastatic melanoma with sinusitis alone or in combination with other irAEs (OS_ICI_ of patients with advanced or metastatic melanoma and sinusitis alone was 44.7 months, 95% CI 25.2–64.2, vs. 28 months, 95% CI 6.8–49.3 for patients with sinusitis with other irAEs, *p* = 0.730). The same applied for patients treated in the adjuvant setting (OS_ICI_ with sinusitis alone was 97.8 months vs. not reached for patients with sinusitis with other irAEs, *p* = 0.114). Again, there was no difference in OS_ICI_ in patients (advanced or metastatic) with rhinosinusitis versus those with other irAEs but without rhinosinusitis (for patients with metastatic disease, OS_ICI_ was 44.7 months, 95% CI 25.2–64.2 vs. 38.8 months, 95% CI 9.5–68.0, respectively, *p* = 0.369, while for patients treated in the adjuvant setting OSICI was 97.8 months vs. NR, respectively, *p* = 0.546). All the above correlations are shown in Figure 3A–F. Among patients treated for metastatic disease, in the univariate analysis, OS_ICI_ was not affected by factors such as age, sex, race, pre-treatment lactate dehydrogenase level, or BRAF status; thus, no multivariate analysis was conducted. Nevertheless, in a Cox regression model comprising rhinosinusitis, eosinophilia, enteritis, and pneumonitis (the latter two being the next most common irAEs), only rhinosinusitis retained its significance as an independent prognostic marker of higher OS_ICI_ (HR, 1.852; 95% CI, 1.049–3.269; *p* = 0.034).

## 4. Discussion

This study represents the largest investigation focused on a homogenous cohort of melanoma patients and the first to evaluate immune-related rhinosinusitis in such a cohort. According to the literature data, immunotherapy may either aggravate or initiate inflammation of paranasal sinuses [12]. The goal of our study was to capture and characterize radiological sinus inflammation occurring after ICI therapy, especially given the emerging understanding of sterile inflammatory responses as a form of irAE. In our population, none of the patients had imaging findings compatible with rhinosinusitis prior to ICI administration.

An interesting and novel finding of this study is the high incidence of rhinosinusitis among patients with melanoma treated with ICIs. In general, the rate and severity of irAEs vary depending on the specific ICI, combination therapies, and patient characteristics. A retrospective analysis reported that 42.1% of patients receiving ICIs experienced irAEs, a lower percentage compared to our cohort, while the most common irAEs were pneumonitis (9.1%), thyroid toxicity (9.1%), cardiotoxicity (8.1%), and dermatological toxicity (6.9%) [14]. A systematic review and network meta-analysis found that the risk of irAEs differs across various ICIs used in advanced melanoma treatment [15]. In this cohort, a frequency pattern of rhinosinusitis based on the administered ICI was not documented. However, since the majority of patients received nivolumab—an ICI regimen typically associated with the lowest risk and most favorable safety profile for irAEs—this regimen was found to be correlated with the majority of rhinosinusitis cases [16].

Notably, as depicted in our results, the prevalence of aseptic and asymptomatic, radiologically confirmed rhinosinusitis is not only quite higher than that described by the FDA database but is probably the most frequent irAE [12]. A possible explanation is that the majority of patients were asymptomatic, and the diagnosis was made mainly through imaging studies. Obviously, in most reported cases, rhinosinusitis represents an aseptic inflammation. Indeed, it has to be noted that symptoms may go unnoticed by both the patient and the physician, either because they are mild and nonspecific or because they may be attributed to other, much more common causes, such as seasonal allergies or upper respiratory tract infections. However, clinicians should remain vigilant for the possibility of bacterial or fungal superinfections, as these complications necessitate thorough evaluation, including imaging and probably culture confirmation. This is particularly critical when immunosuppressive agents are being considered being administered for the management of irAEs, including rhinosinusitis, as the presence of concurrent infections must be carefully ruled out prior to the initiation of such treatments. Another plausible explanation for the observed high incidence of rhinosinusitis could be the specific concentration of T-cells in the paranasal sinuses. A potential correlation may exist between ICI administration and the development of paranasal sinus inflammation, warranting further investigation. Since most patients presented without symptoms, specific therapeutic approaches were not required in this cohort while all symptomatic patients were managed with conservative measures, which proved sufficient. However, the existing literature suggests that immunosuppressive treatments, such as corticosteroids or anti-TNF agents, may be warranted in severe or persistent cases [3,11]. Moreover, in a case report, the anti-IL5-receptor benralizumab was administered to a patient with lung cancer and a past medical history of asthma and nasal polyps, who developed an exacerbation of his eosinophilic asthma with chronic rhinosinusitis with nasal polyposis after nivolumab administration [8]. Regarding severe cases, where conservative measures prove inadequate, symptoms significantly impair quality of life, or sinus polyposis is present, surgical management may be considered [7,10].

We also observed that a statistically significant percentage of patients with eosinophilia developed rhinosinusitis. As reported above, some cases of ICI-related rhinosinusitis are correlated with eosinophilic infiltration in sinus biopsies, suggesting an eosinophilic-driven inflammatory process, while ICIs may also trigger eosinophilic chronic rhinosinusitis, a subtype of sinus inflammation driven by IL-4, IL-5, and IL-13, cytokines that also contribute to peripheral eosinophilia [16]. Emerging evidence suggests that ICIs may provoke a localized Th2-skewed mucosal immune activation, with elevated IL-5 and IL-33 driving eosinophil recruitment—even in asymptomatic or imaging-only presentations. In particular, disruption of the PD-1–PD-L1 axis has been implicated in eosinophilic pneumonia via Th2 cytokines such as IL-4, -5, and -13 [17,18], while eosinophilic enteritis indicates similar gut mucosal activation [19]. Overall eosinophilia during ICI therapy may predict subsequent irAEs [20], and tumor–eosinophil cross-talk (enhanced by IL-5/IL-33) is increasingly recognized as integral to both therapeutic responses and collateral immune adverse events [21,22].

However, more research is needed in order a well-understood causal relationship between ICIs-related rhinosinusitis and eosinophilia to be established.

Further investigation and routine laboratory testing for autoantibodies is likely unnecessary unless there is a clinical suspicion of an underlying multisystem inflammatory disorder, such as granulomatosis with polyangiitis or eosinophilic granulomatosis with polyangiitis—conditions known to affect the sinuses—in which case targeted laboratory investigations should be conducted.

Regarding the time of emergence of rhinosinusitis after ICIs’ administration, similarly with other irAEs, it typically manifests within the first few months of treatment, whereas a small percentage of patients develop rhinosinusitis after a second- or third-line ICI regimen. These findings are in accordance with the current literature, which indicates that some patients may develop irAEs much later, even after multiple cycles of ICIs or upon re-exposure or even after discontinuation of ICIs, emphasizing the need for long-term monitoring of patients undergoing ICI treatment [23,24].

A key strength of the present study is its novel evaluation of the prognostic implications of rhinosinusitis in patients with metastatic melanoma, marking the first such analysis in the literature. The development of this irAE was associated with an improved OS_ICI_ in patients with advanced or metastatic melanoma receiving ICIs, suggesting it could serve as a surrogate marker for treatment response. Moreover, we demonstrated that there was no observed difference in OS_ICI_ between patients with advanced or metastatic melanoma who experienced sinusitis alone and those who had sinusitis combined with other irAEs, while there was also no difference in OS_ICI_ between patients with advanced or metastatic melanoma who had rhinosinusitis and those with other irAEs but without rhinosinusitis. However, in patients treated in the adjuvant setting, rhinosinusitis did not appear to affect OS. Based on the published literature, patients receiving ICIs in the adjuvant or advanced melanoma setting who experience irAEs have been found to present with better prognosis (progression-free survival and overall survival) [25,26]. On the other hand, and similar to our study, Lepper A et al. demonstrated that adjuvant-treated patients showed no significant difference in survival when comparing the irAE versus no-irAE group [27]. The variability in these results likely stems from the distinct underlying mechanisms driving the development of irAEs.

Based on the published literature, the lack of clinical improvement with antibiotic treatment suggests that rhinosinusitis following ICI administration is an irAE rather than a microbial infection. Further supporting this hypothesis, biopsy findings from case reports, though limited, have consistently demonstrated either an eosinophilic infiltration pattern or a T-cell-mediated immune response [6,8]. However, the precise mechanism underlying the development of rhinosinusitis, similar to most irAEs, remains incompletely understood. It is well-established that ICIs enhance immune responses via T-cell activation. As not all patients under ICIs develop irAEs, it seems that different inflammatory pathways may be involved in the development of irAEs, in addition to the so-called endotype/phenotype pathway, which connects underlying pathobiological mechanisms to observable clinical features [28,29,30,31].

This study has certain limitations, including its retrospective design. The number of asymptomatic patients may be overestimated, as the retrospective nature of the study could have led to the omission of mild symptoms during ICI administration due to underreporting, patient recall bias, or lack of documentation. The absence of sinus mucosa biopsies was another barrier in providing valuable insights into the cellular infiltration and underlying pathophysiologic mechanisms. Finally, this study does not investigate whether sinus inflammation resolves and returns to normal levels following the discontinuation of ICIs.

## 5. Conclusions

This study represents the first large-scale evaluation of immune-related rhinosinusitis in melanoma patients treated with ICIs, highlighting its incidence, clinical characteristics, management, and probable prognostic implications in the metastatic setting. We demonstrated a high incidence of a probably underreported irAE, as well as a significant association between the development of rhinosinusitis and the presence of eosinophilia. Regarding prognostic correlations, rhinosinusitis, primarily an aseptic inflammation, was associated with improved overall survival in metastatic melanoma patients, though not in patients treated in the adjuvant setting. Finally, beyond being a significant addition to the existing literature, this manuscript underscores that several irAEs may remain undocumented, necessitating further clinical validation, the keen observation of dedicated physicians, and heightened vigilance.

## Figures and Tables

**Figure 1 cancers-17-02297-f001:**
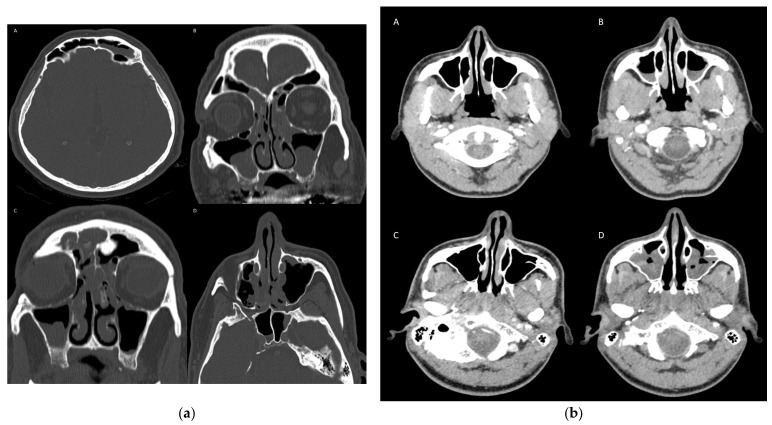
(**a**)**.** Imaging findings of rhinosinusitis. (**A**). Non-contrast CT scan, axial view. Bilateral frontal sinus mucosal thickening, with a left fluid collection in a 77-year-old man with a stage IIIB melanoma under adjuvant treatment with pembrolizumab. (**B**). Non-contrast CT scan, coronal view. Near-total opacification of the ethmoid sinuses in a 68-year-old man with metastatic melanoma under first-line therapy with pembrolizumab. Retention cysts/polyps in both maxillary sinuses, secretions, and mucosal thickening in the right frontal sinus. (**C**). Non-contrast CT scan, coronal view. Total opacification of the right frontal sinus in a 69-year-old man with a stage IIC melanoma under adjuvant treatment with pembrolizumab. Sphenoid sinus and left maxillary sinus mucosal thickening and retention cyst/polyp with fluid collection in the right maxillary sinus. (**D**). Non-contrast CT scan, axial view. Near-total opacification of the ethmoid sinuses in a 72-year-old man with metastatic melanoma under first-line treatment with nivolumab plus ipilimumab. Secretions in the right maxillary sinus, with mild mucosal thickening in both maxillary sinuses. Small collection with an air-fluid level in the right sphenoid sinus compartment, and mucosal thickening in the left. (**b**). Computed tomography imaging findings of the paranasal sinuses before and after initiation of immunotherapy in two patients. (**A**) Pre-treatment axial view demonstrating normal aeration and mucosal thickness of the paranasal sinuses. No evidence of mucosal disease or obstruction is noted. (**B**) Post-treatment axial view obtained after initiation of immunotherapy with fluid collections that partially opacify the maxillary sinuses bilaterally. (**C**) Pre-treatment axial view with no significant findings in the paranasal sinuses. (**D**) Post-treatment axial view showing severe mucosal thickening in the maxillary sinuses bilaterally.

**Figure 2 cancers-17-02297-f002:**
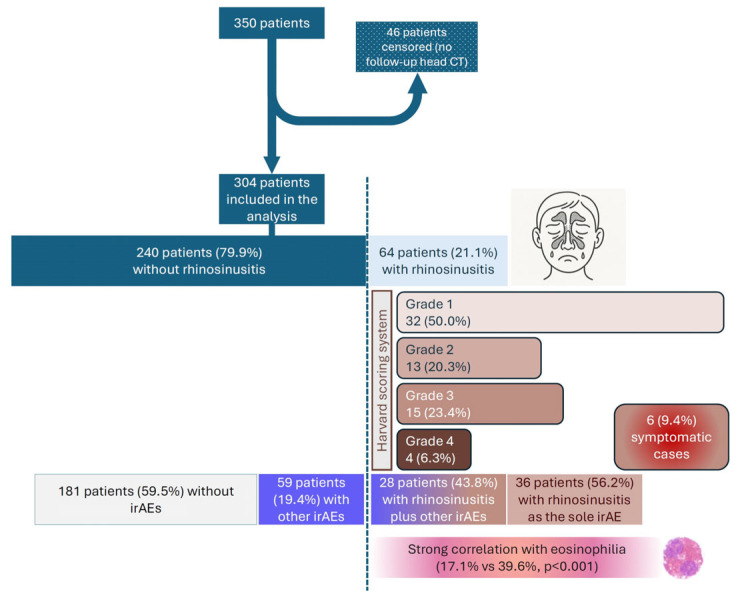
Flowchart summarizing the incidence and characteristics of rhinosinusitis among 304 melanoma patients treated with immunotherapy. Rhinosinusitis was identified in 21.1% of patients, graded by the Harvard Scoring System, with 9.4% being symptomatic. A strong correlation with eosinophilia was observed, and rhinosinusitis occurred both with and without other irAEs.

**Figure 3 cancers-17-02297-f003:**
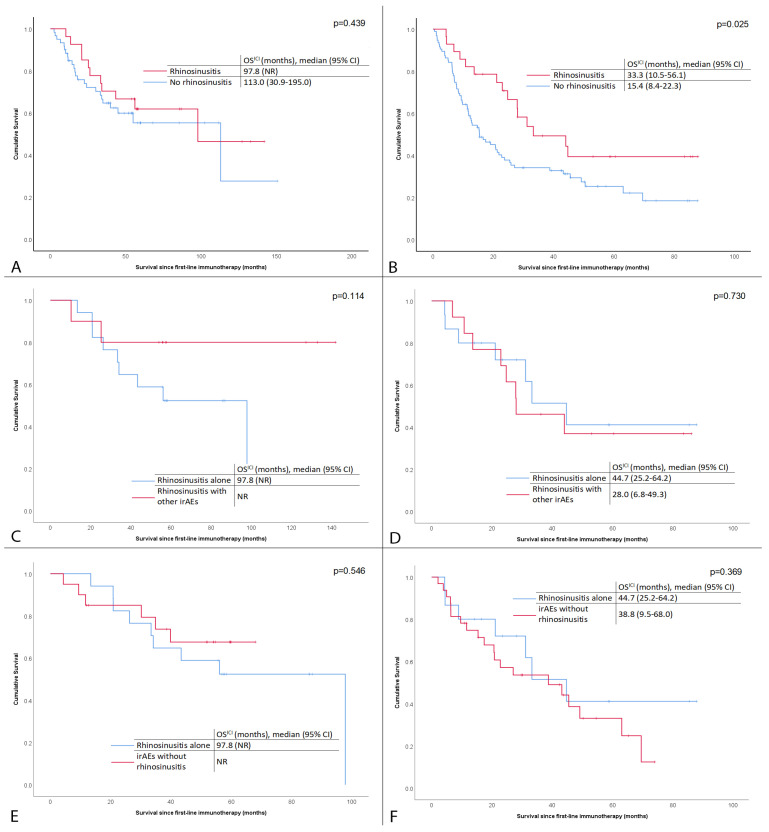
Survival correlations. Kaplan–Meier curves showing the effect of rhinosinusitis on survival after immune checkpoint inhibitor initiation (OS_ICI_) in patients treated in the adjuvant (**A**) or metastatic (**B**) setting with and without rhinosinusitis, in patients treated in the adjuvant (**C**) or metastatic (**D**) setting with rhinosinusitis alone or in combination with other immune-related adverse effects (irAEs), and in patients treated in the adjuvant (**E**) or metastatic (**F**) setting with rhinosinusitis alone or with other irAEs but without rhinosinusitis.

**Table 1 cancers-17-02297-t001:** Current literature data regarding immune-related rhinosinusitis after ICIs administration.

Author, Year	Type of Study, Number of Cases	Type of Malignancy	Type of Treatment	Other irAEs, Management	Sinusitis Related Symptoms	Sinusitis Management	Comments
Dein E, 2017 [5]	Case report, 2	Melanoma	Ipilimumab/nivolumab	1. Enteritis–conjunctivitis–arthritis–urethritis–vitiligo, steroids, anti-TNF 2. Colitis–arthritis–sica syndrome, anti-TNF	1. Maxillary and frontal sinus pain and pressure, without nasal discharge or fever 2. Sinus pressure in the frontal and maxillary sinuses without nasal discharge and fever	Successful management with anti-TNF	Sinusitis with no improvement with antibiotic therapy
Krane NA, 2020 [6]	Case report, 1	Melanoma	Pembrolizumab	-	Headaches, diplopia with lateral gaze, proptotic and chemotic left eye with a left abducens nerve palsy	Voriconazole due to diagnosis of allergic fungal disease without invasive process	Case of allergic fungal rhinosinusitis, a subtype of chronic rhinosinusitis; possibly an ICI-related, T-cell-mediated response to fungal presence; eosinophilic infiltrate with Charcot–Leyden crystals and Aspergillus fumigatus and positive stains for CD-3, a general T-cell marker
Watanabe H, 2020 [7]	Case report, 1	Renal cell carcinoma	Ipilimumab/nivolumab	Eosinophilic airway inflammation, fluticasone furoate/vilanterol trifenatate	Nasal congestion	Fluticasone furoate nasal spray	-
Kassem F, 2021 [8]	Case report, 1	NSCLC	Nivolumab	Peripheral eosinophilia; no further diagnosis was made No treatment	Nasal congestion, rhinorrhea, sneezing, and anosmia	Functional endoscopic sinus surgery and septoplasty	Histopathology with heavy infiltrates of eosinophils
Rembalski S, 2022 [9]	Case report, 1	Lung adenocarcinoma	Nivolumab	Eosinophilic obstructive lung disease, peripheral eosinophilia, benralizumab	Nasal polyposis, anosmia with concomitant bifrontal sinus pressure, obligate mouth breathing	Benralizumab	Antibiotics without benefit
Standiford TC, 2023 [3]	Retrospective 12/108	Melanoma	Pembrolizumab	-	Mucopurulent nasal discharge, nasal obstruction, facial pain/pressure, or decreased sense of smell	Long-term use of saline sinus irrigation (n = 1), steroid nasal spray (n = 1), steroid sinus irrigation (n = 2), and at least one course of antibiotics (n = 3)	None of the patients underwent surgical intervention None of the patients had documented CRS prior to starting pembrolizumab
Hintze JM, 2023 [10]	Case report, 1	NSCLC	Pembrolizumab	Grade 3 nasal polyps bilaterally	Severe bilateral nasal blockage, anterior rhinorrhoea, post-nasal drip, and hyposmia	Topical betamethasone sodium phosphate nasal drops, azelastine hydrochloride, and fluticasone propionate nasal spray	Bilateral functional endoscopic sinus surgery: nasal polypectomy, bilateral uncinectomy, middle meatal antrostomy, anterior and posterior ethmoidectomy, frontal pathway clearance, and wide sphenoidotomy
Tzou-mpa S, 2024 [4,11]	12 Cases	Melanoma	Patients under monotherapy or combination therapy with ICIs	Skin/eye toxicities Endocrinopathies Gastrointestinal toxicities Arthritis ;use of immunosuppressants (corticosteroids, anti-TNF, methotrexate)	Facial pressure Anosmia Rhinorrhea or post-nasal drip Facial pain severe enough to mimic sinus infections	Corticosteroids (topical, inhaled or systemic)/Infliximab/ Methotrexate	Sinusitis with no improvement with antibiotic/antihistaminic therapy (all patients received initially; 2 patients required temporary ICI interruption Nasal mucosal biopsy was performed in one patient; a polypoid lesion with an eosinophilic chorion infiltrate
Pak KY, 2024 [12]	Case report, 1	Endometrial cancer	Pembrolizumab	Peripheral eosinophilia	Nasal obstruction, rhinorrhea, post-nasal drip, and anosmia	Intranasal steroid spray	-

Abbreviations: CRS = chronic rhinosinusitis symptoms; NSCLC = non-small cell lung carcinoma.

## Data Availability

All data are available upon reasonable request.

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
