# Peer review of "Rhinosinusitis as an Immune-Related Adverse Event: Clinical Characteristics, Management, and Prognostic Implications in Metastatic Melanoma Patients"

_cancers, 2025, doi:10.3390/cancers17142297_

Round 1

Reviewer 1 Report

Comments and Suggestions for Authors

I acknowledge that the manuscript explores a niche and relatively under-investigated topic. While the study presents certain limitations in its design and the overall strength of its conclusions may be modest, it is nonetheless a well-written and meticulously conducted work. The authors have approached the subject with scientific rigor and attention to detail, offering valuable insights into an area that has not been widely addressed in the literature. For these reasons, I believe the effort behind the study deserves recognition, and the contribution it makes to the existing body of knowledge should not be overlooked. It would be appropriate for the authors to include pre-treatment CT images of the patients presented in the study, and to compare them with Figure 1. This would help strengthen the statement that none of the patients showed radiological signs of rhinosinusitis prior to the initiation of immunotherapy.

Author Response

Reviewer 1

I acknowledge that the manuscript explores a niche and relatively under-investigated topic. While the study presents certain limitations in its design and the overall strength of its conclusions may be modest, it is nonetheless a well-written and meticulously conducted work. The authors have approached the subject with scientific rigor and attention to detail, offering valuable insights into an area that has not been widely addressed in the literature. For these reasons, I believe the effort behind the study deserves recognition, and the contribution it makes to the existing body of knowledge should not be overlooked. It would be appropriate for the authors to include pre-treatment CT images of the patients presented in the study, and to compare them with Figure 1. This would help strengthen the statement that none of the patients showed radiological signs of rhinosinusitis prior to the initiation of immunotherapy.

Reply

We appreciate the reviewer’s thoughtful suggestion. All pre-treatment CT scans of patients were not available in our institutional imaging archives, as these baseline imaging studies were performed at external radiology centers not affiliated with our institution. As such, we do not have access to all the original CT images, but we did review the radiology reports from these pre-treatment scans, all of which confirmed normal sinus anatomy and the absence of any findings suggestive of rhinosinusitis.

All follow-up CT imaging was conducted within our collaborating center, allowing us to collect and analyze the post-ICI images used in our study and in Figure 1a. Our primary goal in including Figure 1a in the present manuscript was to illustrate the extent and pattern of aseptic inflammatory changes observed after immunotherapy initiation. However we acknowledge the value of direct radiological comparison, so we added Figure 1b which depicts imaging findings of the paranasal sinuses before and after initiation of immunotherapy of two patients with available pretreatment CT-scans in our institution.  

Reviewer 2 Report

Comments and Suggestions for Authors

The authors have developed an interesting topic, and the article is clear, readable, and well-structured. I have a few minor suggestions to improve the manuscript.

In my opinion, the Introduction section should mention the retrospective, multicenter study conducted in melanoma patients receiving immune checkpoint inhibitors (ICIs), focusing on symptomatic immune-mediated sinusitis. Here is the reference for this recent article :
Tzoumpa S, Villette B, Granel-Brocard F, Dutriaux C, Memmi A, Jeudy G, Tafani V, Saint-Jean M, Nardin C, Funck-Brentano E, Corre YL, Quereux G, Maubec E. Symptomatic aseptic sinusitis induced by immune checkpoint inhibitors for metastatic melanoma treatment. Immunotherapy. 2024;16(16-17):1029-1037. doi: 10.1080/1750743X.2024.2399498. Epub 2024 Sep 13. PMID: 39268924; PMCID: PMC11492644.

Regarding Table 1, please indicate whether the author performed a comprehensive review of the literature or not. If not, the manuscript should indicate: “non-comprehensive data”. 

Figure 2 well summarizes the main results of the study. It's well conceived, but I suggest making the text more readable and including a legend, too.

Best regards.

Author Response

Reviewer 2

A. The authors have developed an interesting topic, and the article is clear, readable, and well-structured. I have a few minor suggestions to improve the manuscript. In my opinion, the Introduction section should mention the retrospective, multicenter study conducted in melanoma patients receiving immune checkpoint inhibitors (ICIs), focusing on symptomatic immune-mediated sinusitis. Here is the reference for this recent article: Tzoumpa S, Villette B, Granel-Brocard F, Dutriaux C, Memmi A, Jeudy G, Tafani V, Saint-Jean M, Nardin C, Funck-Brentano E, Corre YL, Quereux G, Maubec E. Symptomatic aseptic sinusitis induced by immune checkpoint inhibitors for metastatic melanoma treatment. Immunotherapy. 2024;16(16-17):1029-1037. doi: 10.1080/1750743X.2024.2399498. Epub 2024 Sep 13. PMID: 39268924; PMCID: PMC11492644.

Reply

Thank you for all these comments and for this valuable reference. This article was mentioned in the section of Introduction and added in the Reference List.

B. Regarding Table 1, please indicate whether the author performed a comprehensive review of the literature or not. If not, the manuscript should indicate: “non-comprehensive data”.

Reply

You are absolutely right, that’s why we made the appropriate changes in the Introduction, reporting the following: Table 1 summarizes the current, limited existing literature after a comprehensive review, addressing rhinosinusitis following immunotherapy in melanoma patients. We also made the relevant changes in the Table 1.

C. Figure 2 well summarizes the main results of the study. It's well conceived, but I suggest making the text more readable and including a legend, too.

Reply

After your suggestion, we made the appropriate changes and we added the following caption in Figure 2: Flowchart summarizing the incidence and characteristics of rhinosinusitis among 304 melanoma patients treated with immunotherapy. Rhinosinusitis was identified in 21.1% of patients, graded by the Harvard Scoring System, with 9.4% being symptomatic. A strong correlation with eosinophilia was observed, and rhinosinusitis occurred both with and without other irAEs.

Reviewer 3 Report

Comments and Suggestions for Authors

This is a well-written and timely retrospective study that explores an underrecognized immune-related adverse event (irAE)—rhinosinusitis—in melanoma patients treated with immune checkpoint inhibitors (ICIs). The large sample size (n=304), use of imaging-based diagnosis, and analysis of survival outcomes are notable strengths. The hypothesis that rhinosinusitis may represent a favorable prognostic marker is both novel and clinically relevant.

Points for Revision:

  1. The manuscript infers that rhinosinusitis may serve as a prognostic biomarker, but causality is not established. This should be more cautiously phrased, especially in the abstract and conclusion (e.g., “may reflect” vs. “is associated with”).
  2. No multivariable analysis (e.g., Cox regression) is provided to control for potential confounding variables that could affect overall survival (e.g., age, stage, other irAEs). This is a major limitation and should be addressed or at least discussed explicitly.
  3. The authors rely solely on imaging (CT findings) for diagnosis. While appropriate, it raises the concern of incidental findings. The specificity of this definition should be acknowledged as a limitation. Sinusitis is a clinical diagnosis and not a radiological one. Only 9.4% of rhinosinusitis cases were symptomatic - can this really be called sinusitis?
  4. The discussion proposes a potential immunologic link via eosinophilia and local immune activation. These are intriguing but speculative and could benefit from references to immunologic mechanisms or ongoing work in mucosal immunology and the link between immune response to cancer and eosonophilia.
  5. Lines 211–213: The assertion that rhinosinusitis may be the most frequent irAE in this cohort needs qualification given the likely asymptomatic and incidental nature of many cases.

Author Response

This is a well-written and timely retrospective study that explores an underrecognized immune-related adverse event (irAE)—rhinosinusitis—in melanoma patients treated with immune checkpoint inhibitors (ICIs). The large sample size (n=304), use of imaging-based diagnosis, and analysis of survival outcomes are notable strengths. The hypothesis that rhinosinusitis may represent a favorable prognostic marker is both novel and clinically relevant.

Points for Revision:

Comment 1: The manuscript infers that rhinosinusitis may serve as a prognostic biomarker, but causality is not established. This should be more cautiously phrased, especially in the abstract and conclusion (e.g., “may reflect” vs. “is associated with”).

Response 1: Thank you for this comment. In the section of Results we report that rhinosinusitis among patients with advanced or metastatic melanoma, was associated with a longer median OSICI (33.3 months 95% CI, 10.5-56.1 vs 15.4 months, 95% CI, 8.4-22.3 months for patients without imaging evidence of rhi-nosinusitis, p=0.025) and that among patients treated in the adjuvant setting, there was no effect of rhinosinusitis on OSICI. Based on these findings, we report in the Abstract that rhinosinusitis may serve as a marker of favorable prognosis in metastatic melanoma patients receiving ICIs. In the section of Conclusions, we made the appropriate changes in order to clarify these findings as follows: This study represents the first large-scale evaluation of immune-related rhinosinusitis in melanoma patients treated with ICIs, highlighting its incidence, clinical characteristics, management, and probable prognostic implications in the metastatic setting.

Comment 2: No multivariable analysis (e.g., Cox regression) is provided to control for potential confounding variables that could affect overall survival (e.g., age, stage, other irAEs). This is a major limitation and should be addressed or at least discussed explicitly.

Response 2: Thank you for the suggestion. This is information that should have been included. In the section of Results we added the following ‘Among patients treated for metastatic disease, in the univariate analysis, OSICI was not affected by factors such as age, sex, race, pretreatment lactate dehydrogenase level, or BRAF status; thus, no multivariate analysis was conducted. Nevertheless, in a Cox-regression model comprising rhinosinusitis, eosinophilia, enteritis, and pneumonitis (the latter two being the next most common irAEs), only rhinosinusitis retained its significance as an independent prognostic marker of higher OSICI (HR, 1.852; 95% CI, 1.049-3.269; p=0.034).

Comment 3: The authors rely solely on imaging (CT findings) for diagnosis. While appropriate, it raises the concern of incidental findings. The specificity of this definition should be acknowledged as a limitation. Sinusitis is a clinical diagnosis and not a radiological one. Only 9.4% of rhinosinusitis cases were symptomatic - can this really be called sinusitis?

Response 3: 

We thank you for this important observation. We totally agree that sinusitis is traditionally a clinical diagnosis and that radiological findings alone may detect incidental or subclinical changes, particularly in asymptomatic individuals. In our study, the term "rhinosinusitis" was applied based on CT criteria using the Harvard scoring system, which has been validated for assessing sinus inflammation radiologically. While only 9.4% of patients met criteria for symptomatic rhinosinusitis, the goal of our study was to capture and characterize radiological sinus inflammation occurring after ICI therapy—especially given the emerging understanding of sterile inflammatory responses as a form of irAE, a comment that was added to the Discussion section. Moreover, this is the reason why we report in the manuscript that this inflammation was aseptic. Finally, and regarding the low rate of symptomatic patients, in the section of the Discussion regarding limitations, we have reported that the number of asymptomatic patients may be overestimated, as the retrospective nature of the study could have led to the omission of mild symptoms during ICI administration due to underreporting, patient recall bias, or lack of documentation.

Comment 4: The discussion proposes a potential immunologic link via eosinophilia and local immune activation. These are intriguing but speculative and could benefit from references to immunologic mechanisms or ongoing work in mucosal immunology and the link between immune response to cancer and eosonophilia.

Response 4: After your valuable comment, we made the following changes in the Discussion: Emerging evidence suggests that ICIs may provoke a localized Th2-skewed mucosal immune activation, with elevated IL‑5 and IL‑33 driving eosinophil recruitment—even in asymptomatic or imaging-only presentations. In particular, disruption of the PD‑1 – PD‑L2 axis has been implicated in eosinophilic pneumonia via Th2 cytokines such as IL‑4,-5,-13 [17,18], while eosinophilic enteritis emphasizes similar gut mucosal activation [19]. Overall, eosinophilia during ICI therapy may predict subsequent irAEs [20] and tumor–eosinophil cross‑talk (enhanced by IL‑5/IL‑33) is increasingly recognized as integral to both therapeutic responses and collateral immune adverse events [21,22]. Due to the aforementioned changes, we added six more References [17-22].

Comment 5:  Lines 211–213: The assertion that rhinosinusitis may be the most frequent irAE in this cohort needs qualification given the likely asymptomatic and incidental nature of many cases.

Response 5: Asymptomatic and aseptic sinusitis following immunotherapy administration represent an irAE that, similar to other irAEs such as hypothyroidism or pneumonitis, may be detected exclusively through serology testing or imaging studies rather than clinical symptoms. This parallels the observation that some irAEs are identified solely via laboratory abnormalities without overt clinical manifestations. We have also reported that radiologically confirmed rhinosinusitis was not reported at baseline imaging studies. In Lines 216-217 we made the following changes: Notably, as depicted in our results, the prevalence of aseptic and asymptomatic, radiologically confirmed rhinosinusitis is not only quite higher than that described by the FDA database, but is probably the most frequent irAE.